# Critical energy landscape of linear soft spheres

**Silvio Franz[1], Antonio Sclocchi[1*] and Pierfrancesco Urbani[2]**

**1** Université Paris-Saclay, CNRS, LPTMS, 91405, Orsay, France
**2** Université Paris-Saclay, CNRS, CEA, Institut de physique théorique,
91191, Gif-sur-Yvette, France

⋆ antonio.sclocchi@universite-paris-saclay.fr

## Abstract

We show that soft spheres interacting with a linear ramp potential when overcompressed beyond the jamming point fall in an amorphous solid phase which is critical, mechanically marginally stable and share many features with the jamming point itself. In the whole phase, the relevant local minima of the potential energy landscape display an isostatic contact network of perfectly touching spheres whose statistics is controlled by an infinite lengthscale. Excitations around such energy minima are non-linear, system spanning, and characterized by a set of non-trivial critical exponents. We perform numerical simulations to measure their values and show that, while they coincide, within numerical precision, with the critical exponents appearing at jamming, the nature of the corresponding excitations is richer. Therefore, linear soft spheres appear as a novel class of finite dimensional systems that self-organize into new, critical, marginally stable, states.

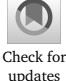
## 1   Introduction

Since more than twenty years, the ideal jamming points of systems of frictionless spheres have shaped our thinking of low temperature glasses, suggested principles underlying amorphous rigidity, and provided mechanisms to rationalize low energy excitations in glasses [1,2]. Topic feature of packings at jamming is mechanical marginal stability. The number of contacts between the spheres is isostatic, in $d$ dimensions each sphere has on average $2d$ contacts, that is the least one for which the system can sustain pressure [3–6]. As such, mechanical marginal stability brings about criticality and diverging lengthscales [7]. The jamming point is a critical point characterized by a set of critical exponents describing both the behavior of bulk physical quantities, such as pressure, energy and contacts [4,8] as well as the microstructure of amorphous packings [9–11]. In particular, a common characterization is provided by local statistics of contact forces and interparticle distances. Marginal stability implies power law behavior of the distribution of these quantities at small argument [9,10] and predicts a non trivial relation between the corresponding exponents [12]. These exponents have been computed exactly in [13, 14] and have been shown to agree -within numerical precision- with numerical simulations of hard and soft spheres in various physical dimensions [15]. The universality class of marginally stable jamming points has been further shown to go beyond finite dimensional sphere systems and to encompass more generally a large class of continuous constraint satisfaction problems in machine learning and computer science [16–19]. For soft constraints, jamming points are isolated critical points: in general, typical soft sphere systems (Harmonic or Hertzian spheres) compressed beyond the jamming point loose most of the salient critical features of jamming, becoming mechanically stable with a finite correlation length. In this paper, we show that if we fine tune the soft sphere interaction potential - choosing it as a linear ramp - we can get a new amorphous solid phase which is mechanically marginally stable and critical for all densities beyond the jamming point. Furthermore, the emerging marginal stability is richer that the one appearing at the boundary jamming transition, with additional system spanning non-linear excitations.

## 2   Model and main results

We consider a set of $N$ spheres in $d$ dimensions whose centers are $d$-dimensional vectors denoted by $\{\mathbf{x}_i\}_{i=1,\dots,N}$. We define a gap between two spheres, say $i$ and $j$, as $h_{ij} = r_{ij} - \sigma_{ij}$, where we have denoted by $\sigma_{ij}$ the sum of the radii of the corresponding spheres and by $r_{ij} = |\mathbf{x}_i - \mathbf{x}_j|$ the distance between their centers. In the overcompressed phase, above jamming, spheres cannot be arranged without creating overlap between them. Therefore one typically defines a pure power interaction potential $v_\alpha(h_{ij}) = f_c (h_{ij})^\alpha_+$ where $x_+ = |x|\theta(-x)$. $f_c$ is a constant that essentially sets the unit of forces. Common choices for the penalty exponent $\alpha$ are $\alpha = 2$ or $\alpha = 5/2$, corresponding respectively to Harmonic and Hertzian spheres. If $\alpha > 1$ the interaction potential is convex and differentiable at $r_{ij} = \sigma_{ij}$, i.e. when spheres just touch. As a consequence, given a contact at jamming, an infinitesimal normal force is enough to

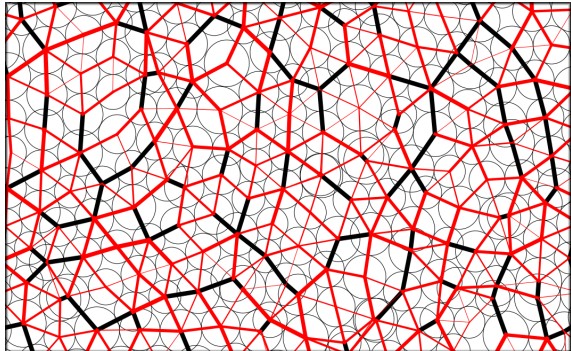

Figure 1: A snapshot of a configuration of linear disks at packing fraction $\phi = 1$. The contact network is in red while the overlap network is in black. The thickness of the lines reflects the intensities of forces. While black lines carry all forces equal to one, red lines, associated to contacts, carry a varying force in the interval $[0, f_c = 1]$.

destabilize it and can cause an overlap between the corresponding particles. Therefore, for $\alpha > 1$ jamming is a singular point in the phase diagram: as soon as the spheres overlap, the system stabilizes. This implies that jamming criticality is washed out when we enter in the overcompressed phase. In this paper, we investigate the potential energy landscape (PEL) of soft spheres interacting through a linear ramp potential, obtained by setting $\alpha = 1$, above the jamming transition point. We show that in this case the jammed phase presents new and unexpected features: the linear ramp potential makes the overcompressed phase critical and marginally stable, characterized by a set of non-linear excitations whose nature is richer than the ones appearing at jamming. The linear ramp potential, which is at the boundary between convex and non-convex interparticle potentials, presents important qualitative differences from the case $\alpha > 1$. First of all, it is non-differentiable: small forces applied to contacts do not necessarily destabilize them. To induce an overlap, a total force greater than $f_c$ is necessary. In addition, the modulus of the force generated by an overlap does not depend on the extent of the overlap itself.

We focus on systems of two and three dimensional polydisperse spheres and produce local minima by gradient descent minimization. Our main findings are:

- Accessible local minima of the PEL are isostatic. Even if there is a finite fraction of overlapping spheres making the total energy finite, there is also an isostatic number of pairs of spheres that just touch. We call *interacting* spheres couples of spheres that either are in perfect contact (*contacts*) $h_{ij} = 0$, or that overlap (*overlaps*) $h_{ij} < 0$.

- Contacts play a crucial role in the stability of the system. Their number is fixed to be exactly equal to the number of degrees of freedom and its fluctuations are suppressed, as it happens at jamming [20]. We show that, as at jamming [21], the spatial fluctuations of the local connectivity of the contact network are hyperuniform implying that the variance of the number of contacts in a volume $V$ grows slower than $|V|$. Conversely, the fluctuations of the number of overlaps follow central limit theorem and spatial fluctuations of the overlap network are only uniform.

- If we look at gap variables and we focus on strictly positive and negative gaps, we find that both distributions have a power law divergence for small argument (in absolute value). The power law exponents controlling the divergence appear to be the same - within numerical precision - for both distributions and very close to the one of positive gaps at jamming.

- Contacts can be associated with forces in the interval $[0, f_c]$. We measure the force empirical distribution and show that it displays two singular pseudogaps, close to zero and close to $f_c$. The pseudogap exponents appear to depend on the packing fraction close to jamming. However, if we carefully separate the contribution of "bucklers", namely spheres that have $d+1$ interacting spheres [11], from the bulk statistics, the pseudogaps are universal and characterized by the same exponents in the whole jammed phase, far from jamming. The values of the critical exponents appear to be the same -within numerical precision - as the one of small force distribution at jamming.

Isostaticity and critical behavior in the force and gap distributions have been shown to appear in the unsatisfiable phase of the spherical perceptron optimization problem with linear cost function, which is a mean field model for linear spheres [22]. The main result of the present work is that these properties appear to survive in a robust way when we go to finite dimension. This implies that jammed packings of linear spheres are characterized by diverging isostatic lengthscales and therefore are critical even far from jamming in the compressed phase. Therefore they provide a new, richer example of self-organized critical, marginally stable, finite dimensional systems.

## 3 Numerical simulations

The linear ramp $v_1(h)$ is a singular interparticle potential and therefore, both for the sake of theoretical comprehension and to perform numerical simulations, it is very useful to smooth the singularity out and to define a differentiable $\epsilon$-regularized potential between particles. From now on we set $f_c = 1$[1]. We can define a regularized interparticle potential as

$$v_1(h; \epsilon) = \begin{cases} 0 & h > \frac{\epsilon}{2} \\ \frac{1}{2\epsilon}(h - \frac{\epsilon}{2})^2 & -\frac{\epsilon}{2} < h < \frac{\epsilon}{2} \\ |h| & h < -\frac{\epsilon}{2} \end{cases}, \tag{1}$$

which in the $\epsilon \to 0$ reduces to the linear ramp potential. The potential energy of a system of $N$ spheres is therefore defined as $H_\epsilon(\underline{x}) = \sum_{i<j} v_1(h_{ij}; \epsilon)$ and we want to study what happens in the $\epsilon \to 0$ limit. In Eq. (1), the non-differentiable point in the origin of $v_1(h)$ has been regularized by an arch of parabola of curvature $1/\epsilon$. Numerically, the regularization enables us to use gradient based routines. Theoretically, the model splits the degeneracy of forces at $h = 0$ in an interval of order $\epsilon$, i.e. $-\epsilon/2 < h < \epsilon/2$; it also allows to properly define the Hessian controlling minima's stability and to argue in favor of isostaticity.

We consider systems of $N$ up to 4096 disks in dimension $d = 2$ and $N$ up to 1024 spheres in dimension $d = 3$, inside a $d-$dimensional box of side-length $L$ with periodic boundary conditions. The particles' radii $R_{\{i=1...N\}}$ are random uniformly distributed between the values $1 - p$ and $1 + p$, with polydispersity $p = 0.2$. The side-length $L$ of the box is set by the volume density $\phi = \sum_{i=1}^{N} k_d R_i^d / L^d$, with $k_d = \pi^{d/2} / \Gamma(1 + d/2)$ where $\Gamma(x)$ is Euler gamma function. Starting from a random configuration of particles' positions, we minimize the energy of the system $H_\epsilon(\underline{x}) = \sum_{i<j} v_1(h_{ij}; \epsilon)$ with the regularized interparticle interaction potential $v_1(h_{ij}; \epsilon)$ defined in eq. (1). The first minimization is run with $\epsilon = 10^{-2}$ using FIRE minimization algorithm [23]. From the obtained configuration, we reduce $\epsilon$ by a factor 2 and repeat the minimization using an approximated conjugate-gradient method (we use the routine L-BFGS [24]). We repeat the procedure halving $\epsilon$ at each step and we stop at $\epsilon \sim 10^{-8}$. Using more accurate minimizers, it is possible to access to lower values of $\epsilon$. We check that

---

[1]Note that $f_c$ sets only the overall scale of the maximal force and therefore we do not loose generality in setting it to one.

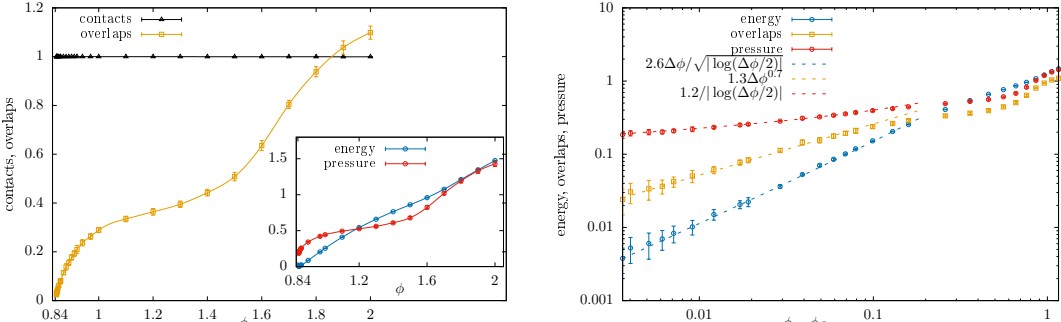

Figure 2: *Left Panel*. Main plot: the isostaticity index defined as $c = C/(Nd − d)$ and the fraction of overlaps defined as $n_O = O/(Nd − d)$ as a function of the packing fraction (we find the jamming transition at $\phi_J \simeq 0.84$). Inset: Behavior of energy and pressure for $\phi > \phi_J$. Energy, pressure and number of overlaps are increasing functions continuous at jamming. Data produced with system size $N = 512$, dimensions $d = 2$, averaged over $\sim 40$ samples for each point. *Right Panel*. The behavior of pressure, energy and overlaps close to the unjamming transition. We attempted some logarithmic fits of the form $e \sim |\Delta\phi|/\sqrt{\log(\Delta\phi/2)}$, $p \sim 1/\sqrt{\log(\Delta\phi/2)}$ and $n_O \sim |\Delta\phi|^{\nu_e}$. The unjamming packing fraction $\phi_J$ is extracted from the fit of the energy.

for the final value of $\epsilon$ we have that $\epsilon \ll \min_{ij}|h_{ij}|$, where $\min_{ij}|h_{ij}|$ is the smallest non-zero gap of the configurations we are looking for. With this procedure we meet the jamming transition at packing fraction $\phi_J^{2d} \simeq 0.84$ and $\phi_J^{3d} \simeq 0.64$. This procedure provides the set $\mathcal{C}$ of the $C = |\mathcal{C}|$ particles pairs $\mu = \langle ij \rangle$, with $i < j$, that are in contact (i.e. $−\epsilon/2 < h_{ij} < \epsilon/2$) and the set $\mathcal{O}$ of the $O = |\mathcal{O}|$ particles pairs $\mu = \langle ij \rangle$ that are overlapping (i.e. that have negative gaps $h_{ij} < −\epsilon/2$). Associated to the contact pairs, there are the scalar contact forces $f_\mu = f_{ij}$ that form a $C$−dimensional vector $\vec{f} = \{f_{ij}\}$, while overlapping spheres exchange forces of intensity 1, whose corresponding $O$−dimensional vector is simply $\vec{1} = [1, ..., 1]$. The scalar contact forces $f_{ij}$ can be computed from the regularized potential of eq. 1 as $f_{ij} = |h_{ij} − \frac{\epsilon}{2}|/\epsilon$, implying $0 < f_{ij} < 1$. Introducing the matrices $\mathcal{S}$ and $\mathcal{T}$, with dimensions $C \times Nd$ and $O \times Nd$ respectively, defined as $\mathcal{S}_{\langle ij \rangle}^{k\alpha} = (\delta_{jk} − \delta_{ik})n_{ij}^\alpha$, with $\langle ij \rangle \in \mathcal{C}$, and $\mathcal{T}_{\langle ij \rangle}^{k\alpha} = (\delta_{jk} − \delta_{ik})n_{ij}^\alpha$, with $\langle ij \rangle \in \mathcal{O}$, where $n_{ij}^\alpha$ is the $\alpha$-component of the versor $n_{ij} = (x_j − x_i)/|\mathbf{x}_j − \mathbf{x}_i|$, we can compute the contact forces $f_{ij}$ also in an algebraic manner using the force-balance condition

$$\mathcal{S}^T \vec{f} = −\mathcal{T}^T \vec{1}. \tag{2}$$

Notice that the system self-organizes in a way that the forces solving the linear system (2) lie in the interval $(0, 1)$.

In Fig.1, we show an example of a configuration we obtain through the numerical procedure just described. In red and black we draw respectively the contact and overlap networks. In the following, we present data for $d = 2$ (number of particles N specified in the captions). The data we got in $d = 3$ are qualitatively similar to the $d = 2$ case and therefore we report them in the appendix.

## 3.1 The jammed phase

In the jammed phase, for $\phi > \phi_J$, particles overlap and therefore the numbers of contacts $C$ and of overlaps $O$, the energy $E$ and the pressure $p$ are different from zero. In two and

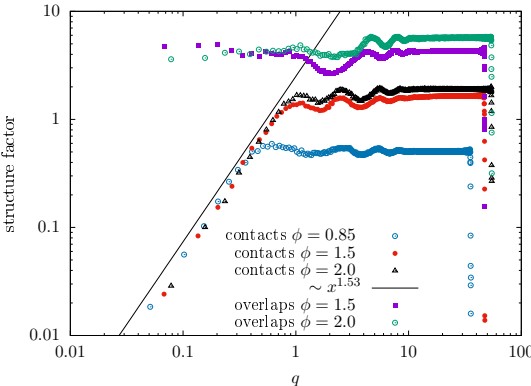

Figure 3: Structure factor of the local connectivity of the network of contacts and overlaps. For small momentum, the structure factor of the contact network decreases down to zero implying hyperuniformity in the fluctuations of connectivity. The exponent controlling the behavior of the structure factor appears to be close to $\sim 1.53$ which is the same found at jamming [21]. On the contrary, the connectivity of the overlap network is not hyperuniform. Data produced with system size $N = 4096$, dimensions $d = 2$, averaged over 44 samples for $\phi = 0.85$, over 50 samples for $\phi = 1.5$ and 48 samples for $\phi = 2$.

in three dimensions, in all the minima we found, once removed the rattlers[2], $C$ is equal to the isostatic value $(N^* - 1)d$, where $N^*$ is the number of spheres which are not rattlers.[3] On the other hand, $O$, $E$ and $p$ are continuous at jamming, having defined the pressure $p$ as $p = V^{-1} \sum_{i<j} |r_{ij}| f_{ij}/d$, where $f_{ij}$ is the force exchanged by the spheres $i$ and $j$ which can be $f_c$ in the case of spheres overlapping or it can be in the interval $(0, f_c = 1)$ in the case of spheres in contact.

This is shown in Fig.2-left where we plot the isostaticity index $c = C/[(N^* - 1)d]$, for a $2d$ system. In the same figure, we plot the overlap fraction $n_{\mathcal{O}} = O/[(N - 1)d]$, and, in the inset, the energy per particle and the pressure. These quantities start from zero at jamming and monotonically increase as the packing fraction grows. We report in Fig. 2-right the behavior of energy, pressure and overlaps close to unjamming trying some preliminary fits. Let us note that the scaling theory developed in [4, 8, 25] becomes just marginal for the linear ramp potential and logarithmic behavior has to be expected. In order to establish the precise form of the scalings close to unjamming one needs to consider proper decompression algorithms that allow to reduce sample to sample fluctuations close to the transition. This goes beyond the scope of this paper and will be the subject of a forthcoming work [26]. To characterize the networks of interaction, we study the fluctuations in the local contact number and overlap number. Following [21], we look at the local connectivity fluctuations of the networks of interactions by measuring the structure factors

$$S_{c,o}(q) = \frac{1}{N} \sum_{i,j=1}^{N} \langle \delta c_i \delta c_j e^{iq \cdot r_{ij}} \rangle , \tag{3}$$

where $c_i$ represents the number of contacts or overlaps of particle $i$ for the contact or overlap structure factors respectively, $\delta c_i = c_i - \langle c \rangle$ is its local fluctuations and the angular brackets

---

[2]Note that rattlers are present close to jamming but their density goes to zero very fast upon entering the jammed phase.

[3]In order to reach a minimum of the system, it is required a minimization algorithm with a spatial resolution of at least $\epsilon$, being $\epsilon$ the regularization parameter: meeting this requirement becomes more challenging when increasing the system size and reducing $\epsilon$.

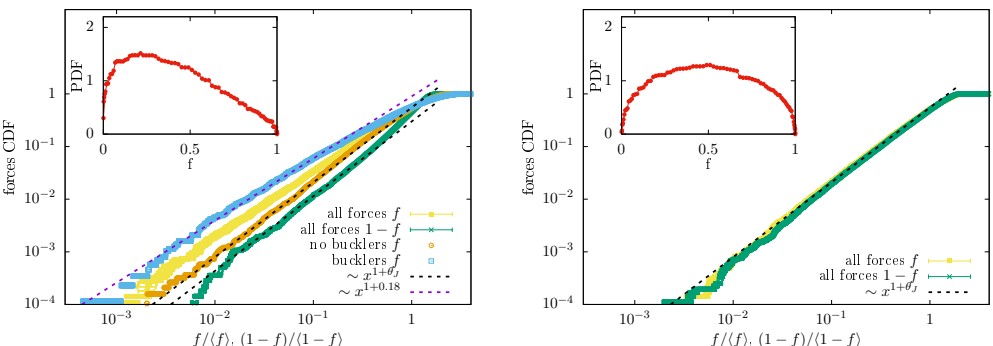

Figure 4: Contact force distributions. *Left panel*: the cumulative of the contact force distribution at $\phi = 0.85$ in $2d$, close to the unjamming transition. We plot the cumulative both starting from the edge at $f = 0$ and at $f = 1$. While a blind statistics of forces is controlled by a hybrid power law exponent, once the effects of bucklers are removed we clearly observe power laws controlled by the mean field exponents, both close to $f = 0^+$ and $f = 1^-$. In the inset we plot the empirical probability distribution function. Data produced with system size $N = 512$, dimensions $d = 2$, averaged over 30 samples. *Right panel*: Cumulative distribution of contact forces close to zero and one at $\phi = 2$ in $2d$, far from jamming. We observe that both distributions follow the mean field exponent. Our statistics is not sufficient to detect any localized excitations at this packing fraction and therefore in this case we consider directly all forces without separating the contribution of bucklers from the analysis. Data produced with system size $N = 512$, dimensions $d = 2$, averaged over 35 samples.

represent average over different minima. We plot both structure factors in Fig.3 for different densities in $2d$. The behavior at small $q$ reveals a different behavior of fluctuations of contact and overlap numbers. The structure factor of the contact network decreases to zero at small argument, while the one of overlaps tends to a positive value. This signals that the fluctuations in contact number are hyperuniform in space, within a volume $V$, the square fluctuations of $C$ scale subextensively in $V$, while the ones of the overlap number are normal and scale as $V$. This difference is a manifestation of the different role that contacts and overlaps play in the stability of the system. As the system is progressively compressed from the jamming point to higher densities, the networks self-organize keeping the number of contacts fixed while increasing the overlaps. As at regular jamming [21], fluctuations of contact numbers away from isostaticity are suppressed and controlled by an infinite lengthscale. We note that the structure factor of the contact network shrinks to zero with a power law that is close to what is observed at jamming [21].

We conclude by noting that while increasing the packing fraction, the fraction of overlaps displays an inflection point around $\phi \sim 1.2$. We empirically observe that at the same point the overlap network seems to undergo to a kind of percolation transition whose nature and properties are left for future investigations.

## 3.2 Statistics of gaps

Having established that the system is isostatic, it is natural to turn the attention to the distribution of non-zero gap variables, which at jamming provides an important characterization of criticality. While at jamming all gaps are positive or zero, here we also have 'negative gaps', quantifying the overlaps between particles. Both the distributions of positive and negative gaps appear to be singular at small argument. In Fig. 5 we plot the cumulative distribution of both positive and negative gaps for several packing fractions beyond the jamming transition

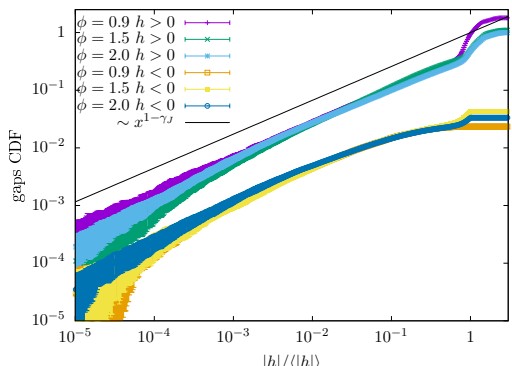

Figure 5: The positive and negative cumulative gap distribution. The $y$-axis of the negative gap is rescaled by an artificial factor 0.1 to improve the readability of the figure. We observe that both cumulative distribution appear to be described by the same power law exponent for small argument. Data produced with system size $N = 512$, dimensions $d = 2$, averaged over 35 samples for densities $\phi = 0.9$ and $\phi = 2.0$ and over 46 samples for density $\phi = 1.5$.

point. The small gap behavior is controlled in both cases by a power law. If we denote by $g_\pm(h)$ the positive and negative gap distribution, we have that

$$g_\pm(h) \sim |h|^{-\gamma_\pm} \tag{4}$$

at small argument. The two exponents coincide within the errors, $\gamma_+ \approx \gamma_-$ and their numerical value appear to be independent of density and equal to the one of positive gaps at jamming $\gamma_\pm = \gamma_J \approx 0.41\ldots$ [13], as predicted by mean field theory [22].

### 3.3 Statistics of forces

Local minima contain an isostatic number of contacts to which we can associate contact forces and study their empirical distribution. Scalar contact forces are naturally defined in the interval $[0, f_c = 1]$ and we observe that, as soon as we enter in the jammed phase, their distribution develops two pseudogaps close to the edges $f \sim 0, 1$ (see Fig. 4). We observe that the exponents controlling the two pseudogaps appear to depend smoothly on the density especially close to jamming and for $f \sim 0^+$. However, following [11] we perform a statistical analysis in which we remove bucklers, namely spheres interacting with $d + 1$ spheres. The result of the analysis is plotted in Fig. 4 and we show that, independently from the packing fraction, the force distribution behaves as

$$p(f) \sim \begin{cases} f^{\theta_-} & f \sim 0^+ \\ (1-f)^{\theta_+} & f \sim 1^- \end{cases}, \tag{5}$$

with $\theta_+ \approx \theta_- \approx \theta_J$, where $\theta_J \simeq 0.42\ldots$ is the critical exponent controlling small forces between hard spheres at jamming [13]. In Fig. 4, we also plot the cumulative distribution function of bucklers' forces close to $f \sim 0$. Again we see a power law behavior controlled - within numerical precision- by the same power law exponents controlling bucklers at jamming of hard spheres [9,11]. Finally, we note that deep in the jammed phase, localized effects such as bucklers (but also rattlers) disappear (with the statistics we have access to) and we do not need to separate them from the statistics of forces to observe a critical power law with mean field exponent $\theta_J$.

# 4 Non-linear marginal stability of linear soft spheres

The exact solution of the perceptron optimization problem with a linear cost function [22] provides a comprehensive mean-field framework that predicts the main features discussed in this paper, namely isostaticity, identity of exponents $\theta_+ = \theta_-$, $\gamma_+ = \gamma_-$, their numerical values and so on. This theory can be adapted straightforwardly to soft linear spheres in infinite dimension [27]. In the next section we complement the mean field theory with marginal stability arguments.

## 4.1 Local stability

The configurations of minima of the PEL at finite energy density contain overlapping particles. It is easy to understand that there should also be pure contacts. The forces corresponding to overlapping particles are constant in modulus (equal to $f_c = 1$) and, without contacts, only very symmetric configurations of particles would be mechanically stable. In fact, more generically, a number of contacts less than $d$ on a particle would require a highly symmetric configuration to be stabilized by only overlaps. The minimal number of contacts necessary to stabilize a single particle is therefore $d$, with a number of overlaps larger or equal to one (or with at least another contact). When we go from the jammed phase towards the jamming point, the number of overlaps vanishes and we recover that at jamming a number of contacts larger or equal to $d+1$ is required to block a sphere. Particles with $d+1$ interactions are prone to local excitations and are usually called bucklers.

## 4.2 The regularized Hessian and isostaticity

Local minima of the linear ramp potential are anharmonic due to the singularity in the pairwise interaction potential. However, one can consider the $\epsilon$-regularized potential and look at the Hessian of local minima in this case. This is indeed well defined and reads

$$\mathcal{H}_{ij}^{ab} = \begin{cases} -\frac{1}{r_{ij}} v_1'(h_{ij};\epsilon)(\delta_{ab} - n_{ij}^a n_{ij}^b) - v_1''(h_{ij};\epsilon) n_{ij}^a n_{ij}^b & i \neq j \\ -\sum_{k \neq i} \mathcal{H}_{ik}^{ab} & i = j \end{cases}, \tag{6}$$

with $n_{ij}^a = (x_i^a - x_j^b)/|\mathbf{x}_i - \mathbf{x}_j|$, $a, b = 1, \dots, d$. Focusing on $i \neq j$, we have the first term, often called prestress, which vanishes at jamming, while we call the second term the elastic term. Because of the regularization, we have that $v_1'(h;\epsilon) = (h - \frac{\epsilon}{2})/\epsilon \, \mathbf{I}[h \in [-\epsilon/2, \epsilon/2]] - \mathbf{1}[h < -\epsilon/2]$ and $v_1''(h;\epsilon) = \mathbf{I}[h \in [-\epsilon/2, \epsilon/2]]/\epsilon$, where we have defined $\mathbf{I}[\mathcal{A}]$ the indicator function which is equal to one if $\mathcal{A}$ is true and zero otherwise. Notice that the Hessian receives contributions both from overlaps and contacts. Overlaps contribute just to the prestress. Contacts instead contribute both to the prestress, with a finite term (notice that $(h - \frac{\epsilon}{2})/\epsilon$ is actually the contact force), and to the elastic part with a term proportional to $1/\epsilon$. This implies that for a variation of the position of the particles such that $|\delta x_i| \lesssim \epsilon$, the energy stored in the elastic term is of order $\epsilon$, and dominates the one stored in the prestress which is of order $\epsilon^2$. This is a crucial property, which is at the basis of isostaticity and the criticality of non-linear excitations in the compressed phase.

Despite giving only a relatively small contribution, the contribution of the prestress term is important. In fact, as usual in repulsive sphere systems, this is a destabilizing term (it corresponds to a negative definite matrix) that, though small, would imply unstable directions if the elastic part is not full ranked. We conclude that the number of contacts should be at least isostatic so that the total matrix is positive definite and the minimum is stable.

The Hessian is therefore dominated by its isostatic random elastic part. Isostatic random matrices are gapless [4, 17, 28–31] and characterized by an abundance of soft modes, their

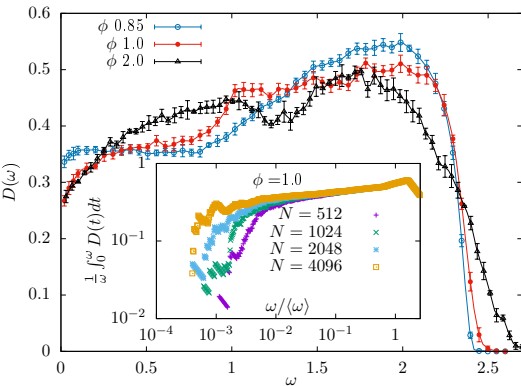

Figure 6: Density of states (DOS) of the elastic part of the Hessian matrix of the regularized potential, see Eq. (6), for different packing fraction above jamming in $d = 2$ and with $N = 4096$, averaged over 44 samples for $\phi = 0.85$, over 42 samples for $\phi = 1.0$, over 48 samples for $\phi = 2.0$. Inset: the finite size behavior of the left tail of the DOS is consistent with having a finite value for $D(\omega = 0)$.

spectrum should behave as $\lambda^{-1/2}$ at small argument, where $\lambda$ represents the eigenvalues. We measure the spectrum of the elastic term matrix, namely the spectrum of $\lim_{\epsilon \to 0} \epsilon \mathcal{H}_{ij}^{ab}$. In Fig.6, we plot the corresponding density of states (DOS) with respect to the vibrational frequency $\omega = \sqrt{\lambda}$. Varying the density from $\phi = 0.85$ to $\phi = 2.0$, our numerical simulations are compatible with having a constant DOS for $\omega \to 0$. In the appendix we develop a mean field theory for such behavior supporting this numerical finding.

## 4.3 Non-linear excitations

Further information can be gained considering a non-linear stability analysis for the local minima of the PEL. The data we presented clearly shows that minima are isostatic configurations where the distributions of both positive and negative gaps display a power law behavior at small argument. At the same time the isostatic delta peak of marginally satisfied gaps is accompanied by a contact force distribution which has two pseudogaps close to forces equal to zero or one. The emergence of these power laws controls the non-linear excitations that dominate the dynamics of the system when perturbing it away from such local minima. One can understand the nature of those excitations generalizing the lines of reasoning employed in [9, 12] for the jamming point. The simplest excitations are the ones in which isostaticity is off by one contact. There are here two possibilities, either separating two spheres in contact and opening a positive gap, or on the contrary pushing two spheres in contact to make them overlap and create a negative gap. The softest excitations are then the ones corresponding to either very week contacts in the former case, or to contacts carrying a force close to one in the latter case. When such contacts are removed, the system would become mechanically unstable unless a new contact forms in the system and again we have two possibilities, either a gap closes, or an overlap relaxes to become a contact. Assuming that both processes occur with finite probability, we have $\theta_+ = \theta_-$ and $\gamma_+ = \gamma_-$. Following [9, 12], one arrives at the scaling relation $\gamma_+ = 1/(2 + \theta_+)$ controlling the critical exponents, which is verified by both our numerics and the mean field theory of [22].

## 5 Discussion and Conclusion

In this work we have described the emergence of a new critical phase obtained when linear spheres are compressed above the jamming point. The criticality of local minima of the PEL of linear soft spheres is described by a set of power laws controlling the positive and negative gap distributions as well as contact forces. The critical exponents controlling such distributions appear to be numerically indistinguishable from the corresponding ones at jamming. Furthermore, the critical behavior is again directly controlled by isostaticity of local minima. This is an interesting result that opens the way to study jamming criticality in a different and complementary way. Indeed, typically, in order to look for the critical properties of the jamming transition, one needs to fine-tune the numerical simulations in order to be close to jamming. Linear soft spheres instead allow us to get to jamming-like critical configurations just by looking at local energy minima which can be obtained using standard numerical routines to minimize the energy. In this case, no fine-tuning is needed. The rich physics that we observe in linear spheres is due to isostaticity which we robustly find with descent dynamics in local minima at finite $N$ [20]. While the relevance of our work for materials as *e.g.* soft colloids or granulars is left for future investigations, the novelty of our results is directly manifested in the emergence of a new mechanism for marginal stability leading to criticality in a finite dimensional system.

Our work opens a series of perspectives: on one hand, it would be extremely interesting to characterize the rheology of strained linear spheres [32,33]. A possible way to look for that would be to perform similar experiments as in Ref. [34–36] and to analyze the statistical properties of avalanches. On the other hand, it would be interesting to investigate other concave penalty exponents $\alpha < 1$, or more complex potentials, to see if different non-linear criticality may arise. Moreover, by switching on temperature, one may investigate if marginal stability emerges at a critical point, the Gardner transition [37–39]. Finally, further work is required to understand the behavior of bulk quantities such as energy and pressure close to unjamming. Likely, this cannot be obtained from the scaling valid for $\alpha > 1$ [4, 8, 25, 40], and it may be important to study the leading corrections to this scaling for $\alpha$ close to one. While our data hints at such phenomenology, further investigations are needed. A possible way to investigate this point would be to progressively compress a configuration from jamming. The dynamics should be dominated by contacts carrying forces close to one becoming overlaps while small gaps becoming contacts with a net flux of gaps from the positive to the negative side of the distribution. How to describe such dynamics is left for future work.

**Funding information**    This work was supported by "Investissements d'Avenir" LabExPALM (ANR-10-LABX-0039-PALM) and by the Simons foundation (grants No. 454941, S. Franz). SF is a member of the Institut Universitaire de France.

## A    Properties of energy minima of linear spheres in three dimensions

Here we report the results of numerical simulations of three dimensional linear soft spheres. We consider $N = 1024$ spheres with varying packing fraction $\phi$. With the polidispersity we are using, the jamming point is at $\phi \simeq 0.64$. We are interested in the properties of the PEL of linear spheres above this packing fraction. As for $d = 2$, we find that our minimization algorithm produces isostatic minima, meaning configurations in which an isostatic number of spheres are perfectly kissing. The properties of the contact network are the same as the ones for two dimensional packings. In Fig. 7-left we plot the dependence of the fraction of contacts with respect to degrees of freedom (computed removing rattlers). In the same plot we also plot

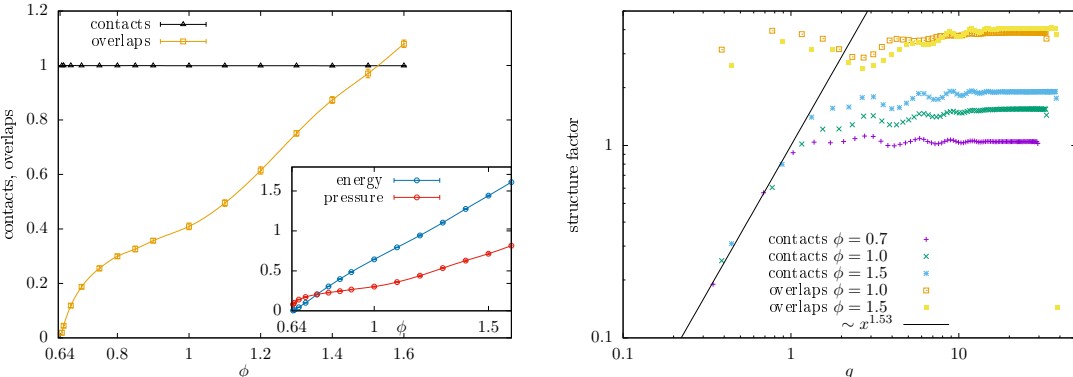

Figure 7: *Left panel*: isostaticity index and fraction of overlaps. While at all densities the minima are isostatic, the number of overlaps increases monotonically. In the inset, the corresponding energy and pressure. *Right panel*: structure factor of the local coordination of the contact and overlap network. While fluctuations of the local connectivity of the overlap network follow the central limit theorem, the ones of the contact network are hyperuniform. Data produced with system size $N = 1024$, dimensions $d = 3$, averaged over $\sim 30$ samples for each density $\phi$.

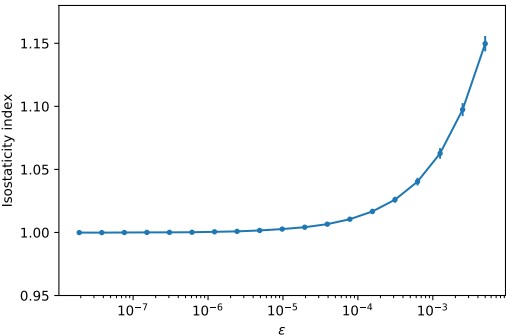

Figure 8: The behavior of the isostaticity index as a function of the regularizer parameter $\epsilon$ for a system of $N = 2048$ spheres in $d = 2$ at $\phi = 2$, averaged over 29 samples. For $\epsilon \to 0$ the system sits in an isostatic minimum.

the fraction of overlaps. While we see that the system is isostatic at all packing fraction above jamming, the density of overlaps increases monotonically. In the inset we report the behavior of the energy and pressure as a function of $\phi$. In the right panel we plot also the structure factor of the local connectivity of the overlap and contact network. We show that while the overlap network obeys central limit theorem, the fluctuations of the number of contacts are suppressed and the structure factor goes to zero for small momenta.

Therefore both in $d = 2$ and $d = 3$ isostaticity is reached when minimizing the energy of the system. In order to see how this happens numerically, in Fig. 8 we plot the isostaticity index as a function of the regularization parameter $\epsilon$ (we plot data for $d = 2$ for simplicity). We clearly see that as soon as the linear potential limit is reached, the system self organizes to sit on an isostatic minimum.

Therefore the main conclusion of this analysis is that, as for the jamming transition, the properties of soft spheres interacting with linear potential do not depend on the dimensionality of the system (apart from local bucklers/rattlers effects).

Finally in Fig.9 we plot the contact force and gap distribution. Contact forces display two

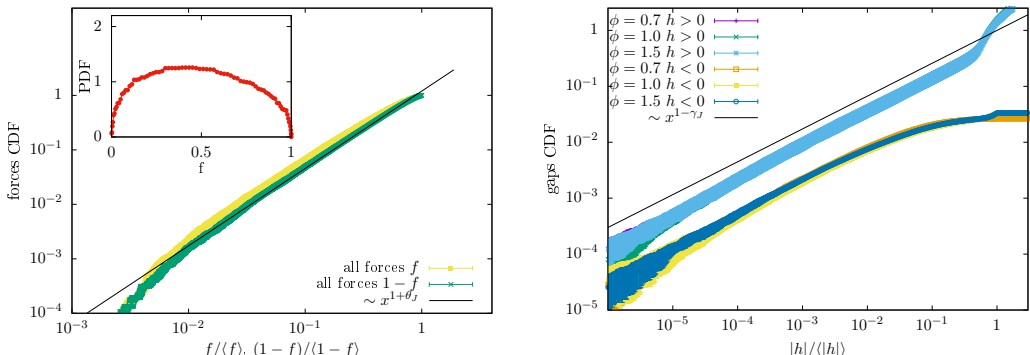

Figure 9: *Left panel*: Cumulative distribution of contact forces close to the two edges at $\phi = 1.5$. The solid line corresponds to the mean field theory prediction. In the inset we plot the corresponding empirical distribution function. *Right panel*: Cumulative distribution of small positive and negative gaps for different packing fractions. The solid line represents the mean field theory prediction. Data produced with system size $N = 1024$, dimensions $d = 3$, averaged over 45 samples for density $\phi = 0.7$, 44 samples for $\phi = 1.5$, 27 samples for $\phi = 2.0$.

pseudogaps close to the edges $f = 0, 1$ while small gaps are controlled by a power law distribution. The critical exponents controlling these distributions appear to be the same (within numerical precision) with the ones of two dimensional linear spheres.

## B Mean field theory of the density of states of the contact network matrix

In the main text we have argued that the elastic part of the contact network matrix has a density of states $D(\omega)$ which goes to a positive constant for $\omega \to 0$ analogously to what happens at jamming. In this section we construct the mean field theory for such behavior. We consider the spherical perceptron optimization problem with linear cost function studied recently in [22], see also [41–43]. This model is in the same universality class of linear soft spheres. It is an optimization problem where a set of $N$ variables $x_i$ arranged in a vector $\underline{x} = \{x_1, \ldots, x_N\}$ lying on the sphere $|\underline{x}|^2 = N$ are sought to minimize the cost function

$$H[\underline{x}] = \sum_{\mu=1}^{\alpha N} |h_\mu| \theta(-h_\mu), \tag{7}$$

where the gaps $h_\mu$ are defined as

$$h_\mu = \frac{1}{\sqrt{N}} \underline{\xi}^\mu \cdot \underline{x} - \sigma, \tag{8}$$

with $\underline{\xi}^\mu = \{\xi_1^\mu, \ldots \xi_N^\mu\}$ a set of $N$ dimensional vectors with components extracted from a Normal distribution and $\sigma$ a constant control parameter. Local minima of the PEL are isostatic meaning that there is an isostatic number of gaps $h_\mu = 0$ and characterized by critical power laws in the gap and forces distribution. The Hessian of the non-analytic minima can be defined by smoothing out the singularity of the linear potential close to $h_\mu = 0$ as we have done with linear soft spheres. Calling $\epsilon$ the smoothing parameter, the Hessian becomes

$$\mathcal{H}_{ij} = \frac{1}{\epsilon N} \sum_{\mu:h_\mu=0} \xi_i^\mu \xi_j^\mu + \zeta \delta_{ij}, \tag{9}$$

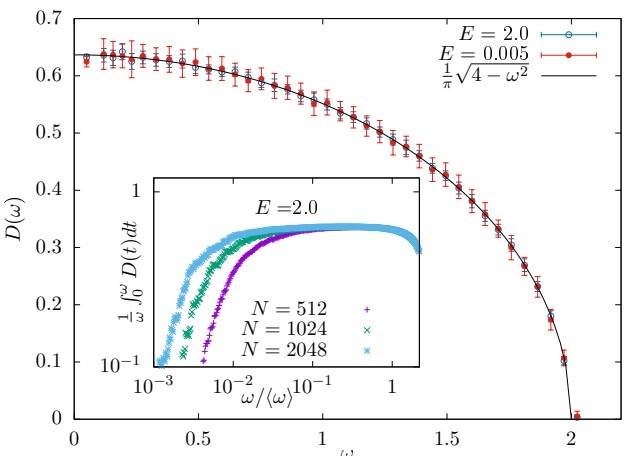

Figure 10: Density of states of the linear perceptron optimization problem for two values of the energy at fixed $\alpha = 5$ (at energy $E = 2$ averaged over 30 samples with system size $N = 2048$, at energy $E = 0.005$ averaged over 50 samples with system size $N = 1024$). The black line is the theoretical prediction given by Eq. (10). In the Inset we plot the left tail of the DOS for different sizes which shows that the behavior is compatible with having a positive DOS for $\omega \to 0$.

being $\zeta$ a Lagrange multiplier needed to enforce the spherical constraint which plays here the same role of the prestress in spheres. In the glassy phase, $\zeta < 0$ and therefore for $\epsilon \to 0$ one needs to have isostatic minima. Since the system is isostatic, assuming that the patterns are random and the only ingredient that matters for the statistics of the Hessian is the number of contacts [17], we get that the spectrum of the Hessian is a given by a Marcenko-Pastur distribution for the eigenfrequencies $\omega$ given by

$$D(\omega) = \frac{1}{\pi}\sqrt{4 - \omega^2}. \tag{10}$$

This result holds in the whole glassy phase regardless of the energy, being the glassy phase always isostatic. In Fig. 10 we plot the density of states as extracted from numerical simulation for $\alpha = 5$ and at two different values of the energy and we compare it with the theoretical prediction.

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
