# Peer review of "Critical energy landscape of linear soft spheres"

_SciPost Physics, doi:SciPost Phys. 9, 012 (2020)_

## Round 2 · Referee Report · Anonymous (Referee 1) · 2020-5-4

Report

Guided by previous mean-field results, the authors study jamming in soft 2D and 3D particle packings where the interparticle potential is a linear ramp. Their main finding is that these systems are critical in a whole interval of packing fractions. This is a very relevant result since, for example, it suggests that mean-field theory (MFT) could be quantitatively accurate also in low dimensions on more than a null-measure region. More implications are clearly described by the authors in the conclusions.
The results are supported via a number of numerical observations on the behavior of these packings. Among those, the fact that they are isostatic and hyperuniform even in the jammed phase. In overcompressed packings, this is possible due to the distinction between kissing and overlapping particles. The indication that correlation lengths are diverging in a whole region of overcompressed packing fractions is evidence for criticality, and provides strong connections to MFT. These connections are explored through the power law scalings of forces and gaps, which are consistent with those found in the related mean-field model (the linear perceptron). Also the density of states is compatible with a finite density of zero modes in the thermodynamics limit, for many packing fractions.
The numerical simulations are performed on smaller systems than in equivalent studies on Harmonic or Hertzian packings, but this may be attributed to the epsilon smoothing that is imposed to the Hamiltonian, which requires to perform several rounds of minimization in order to send epsilon to zero. This does not influence the principal claims of the paper, since an accurate quantitative analysis of scalings is left to future work (it would however be nice to see figure 2 in a log scale, to get an intuition on e.g. the power law $p\sim(\phi-\phi_c)^\beta$ ). The results are given mainly in 2D, with less extensive simulations in 3D that are in reasonable agreement. I could not find where the authors report on how many samples were used to obtain their data (this should be reported).
The manuscript is written in a clear and modular manner, that allows for quick consultation of every single finding. In some cases the figure captions do not contain information such as the system size (amend). In figure 4-left, one of the solid curves is almost completely hidden under the points, and the colors of the solid curves are very similar. I suggest putting the lines on top of the data, using different colors, and using dashes.
The authors succeed in providing physical intuition on their results, and spend time relating them with different kinds of approaches used in the literature.
In summary, the results presented in the article are relevant, well supported by evidence, and well presented. I therefore endorse publication in Scipost Physics.

Requested changes

1- Add numerical details 2- Some captions have missing information on the depicted data 3- It would be nice to see figure 2 in a log scale, to get an intuition on e.g. the power law $p\sim(\phi-\phi_c)^\beta$ 4- Improve readability of figure 4

---

## Round 2 · Referee Report · Anonymous (Referee 2) · 2020-5-28

Report

The authors study a system of soft spheres, in 2 and 3 dimensions, at and beyond the jamming point. Beyond jamming, pairs of spheres interact through a linear ramp potential, which depends on the degree of overlap (negative gap) of the pair of spheres. The linear potential, being singular at zero gap, introduces a series of peculiar properties on the overcompressed amorphous state beyond the jamming point. It is found that local minima of the landscape are isostatic, marginally stable. Then, it is possible to characterize the overcompressed phase from the properties of local minima of the landscape. After regularizing the singularity of the potential and proceeding to numerical minimizations, the authors show a series of interesting properties of these kind of systems: the whole jammed phase is critical, i.e. beyond the critical packing volume the system self-organizes in a critical state. The distribution of local contact forces and the gap distribution show power law behaviour, with exponents compatible with those at the jamming point. While it is not known if such peculiar behaviour can be seen in realistic systems, the presence of a whole critical phase is interesting in itself. The authors also analyze the nature of non-linear excitations, finding interesting pseudogap behaviour at the extremes of the contact force spectrum.

I found the work interesting, it is well written and describes novel behaviour related to the jamming phenomenology. I have some minor questions, mainly of informative character to the reader, that I suggest the authors to consider including in a final version:

1- How is the jamming transition identified from the properties of locally stable states of the landscape. I mean, what is the change in the stability of these states on approaching the jamming transition ?

2- In page 2, second item, it is said "the spatial fluctuations of the local connectivity of the contact network are hyperuniform''. Although references to previous literature are given, I would find useful to define what "hyperuniform'' means in this context.

3- At the end of Section II, subsection B the authors say "We conclude by noting that while increasing the packing fraction, the fraction of overlaps display a plateau around $\phi \sim 1.5$''. I would say that, if a plateau exists at all in the curve shown if Figure 2, it covers a range between $1.1$ and $1.5$. Do the authors have additional data from say, larger system sizes, to substantiate their claim ?

4- The structure factors of contacts and overlaps are shown in Figure 5. Besides the supression of the structure factor of contacts at large distances, what is the overall interpretation of the behaviour of these quantities. For the same case, why is it constant above a fixed value of the wave vector and what does it mean conceptually ?

5- Finally, the fact that the values of all the critical exponents found are near the mean field ones, both for $d=2$ and $d=3$, implies that mean field universality is generic for finite dimensional jamming systems. Beyond formal statements on the upper critical dimension, what is the physical meaning of this statement in the context of jamming systems ? What is special about criticality in them ?

---

## Round 2 · Referee Report · Anonymous (Referee 3) · 2020-6-2

Strengths

The manuscript succeeds in arguing, based on simulations of static local minimas in the potential energy landscape, that a granular system with a linear repulsive overlap potential between otherwise non-interacting particles have a critical behavior in 2 (and 3) dimensions, very much like in the infinite-dimensional (mean-field) limit. Furthermore, the potential is such that identical critical behavior is achieved for every density above the jamming transition, due to the strange property of the potential that bringing particles closer together does no further increase the force. Particles either don't or do overlap, it doesn't matter how much! Thus, for all pressures, there is always just enough contact (ie, particles sitting at the boundary from 0-to-1 force) for isostaticity, the pressure just balanced by having more or less particles overlapping. Then, the "gap" on both sides, ie, particles just about to touch, or just barely overlapping, satisfies quite symmetric scaling properties. These are found numerically to be identical to the mean field model. These results are quite convincing. And the concern an area of great interest: jamming in soft matter, which seems to have a very low upper critical dimension and thus might be a great link between glassy mean field theory and actual materials.

Weaknesses

Much of the technical detail (ie, the mean-field underpinnings) is deferred to the references, so a bit of knowledge is required to understand the significance of this work. However, focusing on the numerical results and their physical interpretation (and on 2d, leaving 3d for the appendix) is the right decision, as it makes the manuscript very readable.

Report

I think that this is a very well-conceived project, and I believe it should be published in its current form.

---

## Round 3 · Author Response

We would like to warmly thank the referees for their positive feedback on our work. The three reports agree on publishing our work with minor changes. We tried to implement the suggestions from Referee 1 and 3 in both a revisited version of the manuscript and with a point by point reply here below. We also provide the resubmitted manuscript in the Scipost template.

Reply to the questions from Referee 1.

1) We added the numerical details the referee requested. In the caption of each figure (apart the first one) we wrote the system size, number of samples and the dimension of the system.

2) We checked the captions and we fixed them.

3) We thank the referee for raising the interesting question about the critical behavior of bulk quantities (such as the pressure) at the unjamming transition. We are investigating this point extensively but we decided not to discuss it in detail in this work. However, since the referee is asking, we added to Fig.2 a plot close to unjamming where we show that we suspect logarithmic behavior for bulk physical quantities. In order to asses this properly we need to consider a refined algorithm to perform decompressions close to unjamming which allows to reduce sample to sample fluctuations. This will be the subject of a forthcoming paper.

4) We changed the layout of fig.4 and we hope that now it is more readable.

Reply to the questions from Referee 3.

1) Jamming is encountered when the energy of local minima goes to zero. Apart from this property, the scaling description of the unjamming transition induced by the linear ramp potential is still open and we plan to investigate it in a forthcoming paper.

2) The spatial fluctuations of the local connectivity of the contact network are hyperuniform: the variance of the number of contacts in a volume $V$ grows slower than $|V|$. We have explicitly mentioned that in the second item in page 3.

3) We apologize for the confusion, fig. 2 shows an inflection rather than a plateau, we changed the sentence into: "We conclude by noting that while increasing the packing fraction, the fraction of overlaps display an inflection around phi ~ 1.2". For what concerns additional data, we provided in Fig.7 the corresponding figure for 3D packings which shows a milder inflection.

4) The structure factor is the Fourier transform of the space-dependent correlation function. We study its small momentum behavior as a means to access large scale fluctuations. At the large momentum, the structure factor tends to a constant that just describes the amplitude of local fluctuations.

5) Since the independence of the critical exponents from space dimension was first found in jamming many people have been scratching their head in the attempt to understand this property. Although there are proposals, based essentially on finite size scaling analysis, that suggest that the upper critical dimension is 2, how to derive this result form phase transition theory is an open problem. This is due to the fact that we lack a field theoretical description of the phase transition itself. Linear spheres are not an exception and we have the same problem also in this case. It is clear that for jamming phenomenology to take place, a prominent role is played by isostaticity but it is unknown how it could be described at a field theoretical level.

---

## Round 3 · List of Changes

• Added numerical details in the caption of every figure and modified them to make them more readable.
  • Added right panel in figure 2 and discussed it in the main text at page 6-7.
  • Added a sentence in the second item in page 3 to describe briefly the concept of hyperuniformity.

---

## Editorial Decision

published